

# A survey of the impacts of summer droughts in England, 1200-1700

Kathleen Pribyl[1,2]

[1]Climatic Research Unit, School of Environmental Sciences, University of East Anglia, UK
[2]Oeschger Centre for Climate Change Research, University of Bern, Switzerland

**Correspondence:** Kathleen Pribyl (k.pribyl@uea.ac.uk)

**Abstract.** Droughts pose a climatic hazard that can have a profound impacts on past societies. Using documentary sources, this paper studies the occurrence and impacts of spring-summer droughts in pre-industrial England from 1200 to 1700. The types of records, source availability and changes in record keeping over time are described, and an overview of droughts in those 500 years is provided. The focus lies on a structural survey over the drought impacts most relevant to human livelihood.

This includes the agricultural and pastoral sectors of agrarian production, health, the fire risk to settlements and the drop in water levels or dwindling of water supplies. Whereas due the specific characteristics of wheat cultivation in medieval and early modern England, the grain production was comparatively resilient to drought, livestock farming was under threat when rainfall fell noticeably below average. The most important problem in warm and dry summers, however, was the risk to health. Partly steeply raised mortality levels were associated with these conditions during the study period, because malaria, gastrointestinal

disease and plague showed an affinity to heat and drought. Adaptation strategies to reduce the stress posed by summer droughts are included in the study.

## 1   Introduction

The occurrence of drought across the British Isles has attracted the attention of meteorologists since the nineteenth century. G.J. Symons' work on hydrometeorology is well known (1887). He was not only studying the rainfall patterns of his own time by

establishing an extensive rain gauge network across the British Isles (Jones, et al. 2007), but also investigated the occurrences of droughts over the past on the occasion of the 1887 drought (Symons, 1887). Brooks and Glasspoole (1928) based their work on past droughts on the catalogue of Symons. After them most research has been done focused on the instrumental period, as by Jones et al. (1997), Briffa et al. (2009), Cole and Marsh (2006) and Marsh et al. (2007). Recently Ireland's drought history was investigated by Wilby et al. (2016), Murphy et al. (2017) and Noone et al. (2017). Some of these works also

include documentary data which is used for an evaluation of the drought conditions and impact research (Cole and Marsh, 2006; Murphy et al., 2017; Noone et al., 2017).

Drought in the pre-industrial period has received comparatively little attention. Using information in the form of direct weather references or proxy data in documentary sources, drought occurrence in England was included in the studies by Jones et al. (1984), Ogilvie and Farmer (1997), Pribyl (2017) and Pribyl and Cornes (2019a,b), and for Ireland by Dooge (1985). Over

the last decades a new source for the study of droughts on the British Isles extending back to Antiquity has become available: precipitation reconstructions based on tree-ring data (Cooper et al., 2012; Rinne et al., 2013; Wilson et al., 2013). These data



have been integrated with more data coming from continental Europe and gridded spatial fields have been produced in the Old World Drought Atlas (OWDA) by Cook et al. (2015) covering the last two millennia. However, the uncertainty connected with tree-ring based reconstructions of precipitation as indicated by Bothe et al. (2019) highlights the value of using documentary
sources to verify the occurrence of historic droughts.

Drought as a hazard was rarely in the focus of economic or agricultural historians, because on the British Isles severe harvest failures are mostly linked to excess rainfall. Hence, from Hoskins (1964, 1968) to Campbell (2009, 2016) research attention was concentrated on the impacts of wet extremes on agricultural output. In pre-industrial times such precipitation-induced harvest failures were the most common cause of subsistence crises. In their study of drought hazards in Ireland Murphy et al. (2017)
could even go so far as to state that the words 'Irish drought' may be perceived as an oxymoron, even though drought does indeed pose a risk to modern day Irish agriculture. In recent years drought impacts have moved more into the focus of research of historians and meteorologists alike. Cole and Marsh (2006) included impacts in their survey over droughts in England and Wales after 1800, Stone (2014) investigated the subject in England for the 1320s and 1330s; Pribyl (2017) considered droughts and their impacts from the late thirteenth to the end of the fifteenth century, Pribyl and Cornes (2019b) gave a general overview
over drought impacts in medieval and early modern England, and Murphy (2017) focused on Irish nineteenth-century droughts.

Trenberth et al. (2014) found that under the conditions of global warming, some studies point to an increase of droughts over Europe, whereas other studies indicated a decrease. Hence more research into past droughts is needed. Briffa et al. (2009) found an increase of droughts in Europe, especially over Central Europe; this increase was mainly due to higher temperatures in the years around 2000. In Ireland, fewer droughts were recorded over recent decades (Murphy et al., 2017), and even though
droughts have affected England over this time, in popular memory they are still overshadowed by the drought 1976; in general no increase of droughts could be noted before the middle of the first decade of the twenty-first century (Cole and Marsh, 2006).

This paper studies summer drought occurrence and impacts in the centuries between 1200 and 1700 in pre-industrial England. It gives a short overview and examples of past droughts, but does not constitute a catalogue; it rather attempts a structural survey of drought impacts on the most important fields of human life: the food supply – i.e. the agricultural and pastoral sectors
– and health, which could be under threat by the outbreak of diseases or the decreased access to clean drinking water. The energy supply for grain or industrial mills, the availability of river transport, the fire risk and adaptation strategies are also considered.

## 2   The documentary sources

The documentary sources used in this study come mainly from southern and eastern England (Fig. 1). Not only are these
regions more liable to drought than northern or western England, but for most of the study period they dominate the historical record. Before 1200 weather information exists, going even back into the Early Middle Ages, but it is at the beginning of the thirteenth century, that the frequency and nature of the information permit a quantitative analysis (Ogilvie and Farmer, 1997). This paper focuses on droughts occurring in the summer half year; droughts in this time of the year interfered with growing crops and herbage. Whereas severe droughts caught also the attention by townspeople, summer seasons with dry periods (but





not severe droughts) were more likely to be recorded in a rural setting by people involved in agriculture. The definition of seasons shifted over time in England. Medieval astronomers were already familiar with the division of the year based on the sun, but in general, in the Middle Ages, in particular for people in the countryside, summer began, when the growing season got into full swing, so at a variable date in April or May. It lasted until the harvest season, autumn, in August and September. The end of the harvest season was marked by Michaelmas, 29 September, but winter was considered to start with the onset of

wintry weather. Spring was perceived as an event rather than a season (Ogilvie and Farmer, 1997; Titow, 1960).

The documentary sources used in this paper are mainly of narrative nature, which means they primarily record extreme weather and its impacts; a sample of drought descriptions is provided in Table S1 in the online Supplementary Information. For the centuries up to 1600 the narrative sources are mostly chronicles, first written by monks, later by lay persons from an urban background. For the sixteenth and seventeenth century parish records and the Venetian State Papers provide information

and after c. 1620 the keeping of personal and farming diaries became popular. Administrative sources, such as manorial or municipal accounts also provide direct and indirect information on past droughts, indirect information also reflect normal conditions. For the Middle Ages many manorial accounts survive, these supply data for agricultural and pastoral farming and often mention weather conditions inferring with these activities (Brandon, 1971; Pribyl, 2017; Stern, 2000; Titow, 1960, 1970). Using the narrative sources and the direct weather references from manorial accounts, Ogilvie and Farmer (1997) constructed

monthly indices for temperature and precipitation for the period 1200 to 1430.

Manorial accounts also often give information on the harvest length which is an indicator of the rainfall frequency and amount shortly before and during harvest; for East Anglia this information has been collated into a precipitation index for the months July-September from 1256 to the mid-fifteenth century (Pribyl, 2017). After c. 1290 the data are almost continuous, they help to identify extremes and are particularly apt at displaying interannual variation. Averaged over 11-years the precipitation

index follows the trends of tree-ring based precipitation reconstructions from Norfolk and Hampshire – even though temporal overlap of the precipitation index (July-September) and the tree growing season (March/May-July) is limited (Fig. 2) (Brassley et al., 1988; Cooper et al., 2012; Jones and Baker, 1964; Pribyl, 2017; Wilson et al., 2013). The pivotal importance of rainfall conditions for the grain harvest is demonstrated by John Crakanthorp counting the rain days per harvest in his harvest diary 1682-89 (Brassley et al., 1988); essentially wet weather at harvest prolongs the drying time of the grain crop and wet grain is

liable to spoil and loose nutritional value.

## 3 Characteristics of drought in England

In England short droughts occur most commonly over spring and early summer. The start of spring is in general dry; the 'drought of March' enabled the ploughing and sowing of the spring grains, barley and oats (Whitley, 1850; Daley, 1970), but under drought conditions very low levels of rainfall then continue to mark the months until about July. Frequently the drought

then comes to an end and in August, at harvest time, higher rainfall levels can be expected (Whitley, 1850). A damp autumn, as is typical in England, reduces the built-up drought pressure (Jones and Baker, 1964), however a run of such dry seasons can nonetheless deplete ground water levels and cause a hydrological drought.





Jones et al. (1997) classified English droughts into short droughts of up to nine months length and ending in autumn, or multiannual droughts of 15-18 months duration containing in general two summers and one winter and having their greatest impact on the south-eastern part of Britain. Indeed dry years frequently cluster between 1800 and 2006 according to Marsh and Cole (2007). The following section on droughts throughout the centuries demonstrates, that this pattern also applies to the medieval and early modern droughts. For the formation of multi-annual droughts dry winters are also important, and will therefore be shortly considered for the pre-industrial period.

Marsh and Cole (2007) also identify long droughts, i.e. decades or even longer, when repeated periods of severe rainfall deficiency occur. In the documentary sources before 1700 the focus lies on the intensive drought periods, hence mostly single-year droughts and short clusters are described, whereas long droughts do not appear as such, probably also due to a certain level of acclimatization. Nonetheless on occasions the occurrence of long droughts can be deduced from the historical record. The 1240s and early 1250s, the first decade of the 1300s, 1325 to c. 1340, the second half of the 1380s, the 1410s, the 1470s, the 1610s, the 1630s, parts of the 1660s, 1670s and 1680s seem to fit this pattern.

## 4 Droughts over the centuries

In the thirteenth century there is evidence in the documentary sources for prolonged dry conditions in spring and summer 1222, 1232, and in the mid-1230s (especially 1236, see Table S1). Drought also reigned during most of the 1240s, in the first half of the 1250s, in the early 1260s, in 1272 and the second half of the 1270s, from the mid-1280s to 1291 and in the last years of the century.

Probably the best described drought of the thirteenth century came in 1252. Matthew Paris, a monk at St Albans, dedicated three paragraphs to it (see Table S1). Between March and July it was dry and from April onwards also very warm. The lack of moisture impacted on the sown grain crops, and as the drought drew on, the landscape became parched, pasture was lacking, the cattle was wasting, the foliage was withered and the fruit and nut harvests lost, the soil cracked open and even the birds suffered although there was a multitude of flies. Later in the summer a cattle murrain broke out and the people fell victim to disease. The problems encountered on the manors of the Bishopric of Winchester, which were situated in southern England centering on Hampshire, clearly fit this pattern: the ploughs broke on the hard soil, pea crops were lost, water levels dropped in the wells and pasture fell short; on one manor spring ploughing was impossible for six weeks (Titow, 1960). The drought and heat stretched across the Irish Sea to Ireland, where again water levels were low, trees burnt up in the sun and the grain harvest was very early (Hennessy, 1871). At harvest time rain came on. The manorial accounts from the Bishopric of Winchester indicate that already summer 1251 had seen a dry period, and in 1253 another dry spring and early summer followed, to be repeated in 1255 (Luard, 1880; Titow, 1960). Indeed this pattern was only broken with the year 1256 when the weather turned in mid-August and the very wet conditions, that were to dominate most of the second half of the 1250s began.

During the fourteenth century England experienced drought conditions in the middle of the first decade, in the mid-1320s, the early and later 1330s, the early 1350s, the early 1360s, the mid-1370s, the mid- and later 1380s and early 1390s. The drought in the mid-1320s and early 1330s is well documented and its impact on farming has recently been studied in detail





(Anon., 1882; Aungier, 1844; Clyn, 1849; Pribyl, 2017; Pribyl and Cornes, 2019a; Stone, 2014; Titow, 1960). In the post-Black Death environment fewer narrative sources are available for England, but in 1361 John of Reading and the Pipe Roll of the Bishopric of Winchester describe an exceptional drought affecting the country (Titow, 1970; see Table S1). Grass and hay production suffered in particular; the drought was then followed by a 'second spring' in October. The Winchester accounts also

explicitly refer to heat, which is unusual for medieval sources. Indeed the spring-summer 1361 had the highest temperature in a reconstruction based on grain harvest dates for the period 1256-1431 (Pribyl et al., 2012; Pribyl, 2017). Grain prices rose in the aftermath of the harvest 1361, and grain exports except to Ireland were forbidden, but the real calamity hitting the country in this year was the Second Pestilence, which killed c. 10% of the English population (Kerling, 1954; Pribyl, 2017).

The number of available written records dealing with weather dwindles further in the fifteenth century, probably reflecting

the persistent negative demographic trend and the tensions of the War of the Roses. Nonetheless it is clear, that the 1410s and the early 1430s were mostly dry. Drought and heat affected the country in 1464, and returned in the early and late 1470s. In 1473 the conditions were extreme and extended over wide parts of Europe (see Table S1). The drought 1464 (see Table S1) coincided with a shortage of currency and the combined effect of these two factors lead to low prices for foodstuffs; another plague wave took hold in the country. Again it was very hot and the spring crops, grass and hay suffered. In the marshland

around Ely insects attacked cattle and people and the death rate rose.

In the sixteenth century the number of available sources augments, particularly in the second half of the century climatic variations are known in greater detail. Up to c. 1550 repeated warm and dry periods took hold, as in the later 1530s. After 1550 their frequency declined, but they still occurred as in 1556 or in the early 1590s. In 1540 an exceptional drought affected large parts of Europe, England included, this summer half year was so extreme as to constitutes a low-probability high-impact

event (Glaser, 2013; Pfister, 2018; Wetter and Pfister, 2013).The London chronicles (see Table S1) all describe a steeply risen mortality across the country due to agues, digestive disease and some plague occurrence. Cattle also died for lack of water, and water courses ran dry, even the Thames was so low as to allow salt water upriver from Old London Bridge. In fact it did not rain between June and late September [Old Style] according to Charles Wriothesley the situation was so severe that Henry VIII as the head of the Church of England ordered the bishops to encourage prayer and processions; in London weekly processions

began in mid-September [Old Style] (see Table S1). The continuator of Robert Fabyan mentions the drought and the ensuing diseases, he also describes the subsequent cold winter 1540-41 (see Table S1).[1]

Over the course of the seventeenth century the available information on weather increases vastly even before the onset of instrumental measurements in the middle of the century: personal and farming diaries, newspapers and notes in parish registers all can contain weather references. In this century droughts are noted in the 1610s, in the mid-1630s, in the early and late

1650s, during the second half of the 1660s and 1670s, in the mid-1680s and then around 1690. The drought in the mid-1630s is already well covered by written records. Winter 1633-34 had, at least in the southwest of the country, been wet, but then the following spring was very dry and until midwinter there was such a rainfall deficit that springs and brooks failed and cattle needed to be driven over long distances to water (see Table S1). The drought affected the rest of southern England and East

---

[1] In should be noted, that most of the references to 1540 in English weather compilations such as Lowe's 'Natural phenomena and chronology of the seasons' (1870) actually stem from Swiss sources and the English information has been erroneously dated to 1541 (Pribyl and Cornes, in 2019a).



Anglia (Jones et al., 1984). The following winter was very cold and also long across the country (Blomefield, 1806; Whiteway, 1991). The summer season 1635 was dry again (Hinds, 1921), but the drought and heat then culminated in spring-summer 1636. Plague was present and the Venetian ambassador – used to a southern climate – expected a great scarcity of victuals after a summer drought so severe that even the trees lost their leaves (see Table S1). In Essex then rain came on in August, damaging the harvest, and causing a 'michlemasse spring' of fresh growth in autumn (Webster and Shipps, 2004). Spring 1637, however, saw dry conditions again.

Dry winters are particularly important for the development of hydrological droughts, because the groundwater storage is recharged between late autumn and early spring (Marsh and Cole, 2007). A convergence of cold, dry winters and warm, dry summers indicates an atmospheric blocking action over England; the historical records indicate that such a weather pattern was not infrequent in the past even though it was rare over recent decades. Dry winters are rarely described in the written record, especially before the late sixteenth century. Records do mention dry and cold winters for 1224-25, 1233-34, 1325-26, 1516-17 and 1590-91; at least 1325-26 and 1590-91 were associated with drought conditions in the previous and following summers (Anon., 1865; Fabyan, 1811; Goring and Wake, 1975; Hamilton, 1875; Hewlett, 1889; Jones et al., 1984; Stow and Howes, 1631; Titow, 1960). The dry winter season 1613-14 also saw below average rainfall and fell into a decade of frequent dry summers. Winter 1664-65 was followed by a dry spring (Jones et al., 1984). For the last decades of the seventeenth century Anthony Wood, an antiquarian in Oxford, provides a record of dry cold winters: 1669-70 after the dry summer 1669, 1675-76 lying between two drought summers and winter 1678-79 occurred between two spring-seasons which saw dry periods (Clark, 1892). This pattern was maintained for 1684-85, a severe winter on the whole, hardly any precipitation fell throughout February and most of March as Wood and Robert Boyle noted (Clark, 1894; Cornes, 2019). The dry 1684 was already preceded by a dry cold winter in 1683-84 (Dobson, 1906; Gill and Guilford, 1910; Jones et al., 1984).

Dry winters are, however, normally cold winters, and cold winters are more frequently mentioned in the written records than dry winters, also before the late sixteenth century. Clustering of cold winters and dry summers during the study period was not untypical. Examples are the following cold winters, which were framed by two dry summers: 1247-48, 1253-54, 1304-05, 1305-06, 1332-33, 1338-39, 1339-40, 1413-14 and 1634-35 (Hewlett, 1889; Luard, 1876, 1880; Pribyl, 2017; Titow, 1960, 1970; Whiteway, 1991). Severe winters before dry summers are recorded for 1224-25, 1267-68, 1327-28, 1422-23 (Anon., 1865; Pribyl, 2017; Titow, 1960, 1970). The hard winter 1434-35 was most likely preceded by an extremely dry summer as indicated by the OWDA – but the stream of documentary sources is almost breaking up for England in the decades of the mid-fifteenth century and hence no reference to the drought 1434 survives – and summer 1435 was probably also dry (Anon., 1880b; Gregory, 1876; Harris, 1827; Langdon, 2004; Titow, 1970).[2] Hard winters after summer droughts include 1225-26, 1268-69, 1288-89, 1291-92, 1309-10, 1329-30, 1358-59, 1407-08, 1416-17, 1464-65 and 1540-41 (Anon., 1866a, 1880a, b; Fabyan, 1811; Haydon, 1863; Herryson, 1840; Pribyl, 2017; Riley, 1864; Titow, 1960, 1970). In general during the dry decade of the 1410s, the cold and partly long winters 1409-10, 1412-13, 1413-14, 1416-17 and 1418-19 were preceded or followed by dry summer half years (Pribyl, 2017; Titow, 1970). Many of the hard winters between 1209 and c. 1450 are known from the

---

[2]The mill at Birdbrook, Essex, stopped working due to a lack of water in the accounting year 1435-36, (Langdon, 2004). The year 1436 is generally known to have been wet over Europe (Alexandre, 1987), thus autumn 1435 is more likely for the mill to have stopped working.





Pipe Rolls of the Bishopric of Winchester and the vast majority of these winters were associated with dry summer seasons, a few with unexceptional summers and none with wet summers (Titow, 1960, 1970).

## 5  Drought impacts

### 5.1  Agriculture

Even when considering only this small sample of drought descriptions and the references in Table S1, the most important concerns of a pre-industrial society facing drought already become clear: agricultural and pastoral productivity, health, falling water levels and their implications. In 1636 the Venetian ambassador expected scarcity to manifest itself in the foreseeable future and grain crops of stunted growth or even non-germinating seed corn appear in sources for 1236, 1252 and 1361 (Luard, 1880; Tait, 1914; Titow, 1960, 1970); in the latter year grain exports except to Ireland were stopped (Kerling, 1954). However, references to famines are conspicuously absent from the historical record. On occasion even a great abundance of grain is mentioned for harvests in dry summers as for 1287 and 1288 (Anon., 1866a; Ellis, 1859; Rishanger, 1865; Titow, 1960). In 1333 Ireland also enjoyed a plentiful harvest (Hennessy, 1842; Titow, 1960) and in 1464 '[...] whete was worthe iiij d. a busshell, and all maner of vetaille grete chepe [...]' (see Table S1), although this was also connected to a currency shortage at that time, and the same occurred after the harvest 1667 (Macfarlane, 1976).

The narrative and administrative sources reporting on weather conditions, but also works on agriculture in England confirm that indeed 'Drought never bred dearth in England' (Heywood, 1909). This idiom originates in the first half of the sixteenth century, but probably reflects knowledge even older, and it remained common knowledge until at least the late nineteenth century (Inwards, 1994). Even though the optimum growing conditions vary from one cereal to another – wheat, rye, barley and oats – as does their sensitivity to excess wetness or dryness, in the maritime climate of England harvest success is primarily threatened by excess rainfall, and subsistence crises are linked to wet growing and harvest seasons. None of the corn crops thrives in a wet summer, even though barley and oats bear such conditions better than wheat (Bowden, 1985). In particular wheat in England during the study period was approaching the limits of its cultivation, it would require more summer warmth and dryness than generally provided in this region (Whitley, 1850; Simpson, 1850). Hence wheat suffered greatly in wet and cool summers, but essentially it was hardly ever too warm and dry for this crop in England (Bowden, 1985; Jones and Baker, 1964; Simpson, 1850; Whitley, 1850). In fact yield data from the extensive estates of the Bishopric of Winchester in southern England between 1209 and 1350 show that almost all outstandingly good harvests (15% above average) occurred under dry conditions: a very dry previous summer and autumn, followed often by a hard winter and then again by a very dry spring-summer (Titow, 1960). Up to about the middle of the seventeenth century wheat was also mostly grown in the English vales, often on undrained clay soils, which increased the resilience of the grain crops to drought and exacerbated the effect of wet weather; the integration of higher ground into cereal cultivation helped to offset the effect excess rainfall on the valleys (Jones and Baker, 1964). However, the comparative resilience of wheat to drought applies even to the conditions of modern agriculture in recent decades, in 1976 the wheat crop fell below average, but it did not fail (Carter, 1978; Martin, 2010).



Indeed the notion that 'Drought never bred dearth in England' can be traced beyond the sixteenth century. Around 1340
William Merle, a fellow at Merton College, Oxford and rector of Digby in Lincolnshire, wrote his 'Tractatus de pronosticacione
aeris',[3] an unusual work on weather forecasting, as it did not focus on using astrometeorological means, as was common at
universities and courts, but highlighted the observation of the 'vulgar' signs of nature such as the flight of birds for short term
predictions. However, the 'Tractatus' is also uncommonly practical in purpose, since it studies the influence of various weather
types on the agricultural production, particularly on the grain crops in an English context. Merle is most likely the first to state,
that dry weather was good for wheat cultivation, and that under drought conditions there could be no famine. This is not only
borne out by the contemporary yield and weather information by Titow for southern England, but also by the English wheat
price. Considering the price in the context of growing season temperature and harvest season precipitation from 1264-1431
demonstrates, that in general warm dry years were associated with low prices and the majority of harvests following on these
meteorological conditions lowered the price further. Warm and dry years also often brought the wheat price back to normal
after famines as in 1318 or later in 1651 and 1652. Sometimes the drought was severe enough to be linked to a moderate
increase in prices, but the occasions when the grain price rose substantially are rare, some of these rises can also be attributed
to other (weather) factors and in no case was a subsistence crisis caused by a warm and dry summer alone (Jones et al., 1984;
Munro, 2008; Pribyl, 2017).

However, drought poses a risk for the spring crops barley and oats and to legumes, and Merle was well aware that these
need humidity after sowing and are vulnerable to prolonged dry weather. When he was writing in the late 1330s he must have
been aware of the impact of drought in the mid-1320s and early 1330s, which was geographically very varied – sometimes
even from village to village, but significant (Stone, 2014). It is the effect that a lack of rainfall has in particular on barley, a
grain that also served as a bread corn in the Middle Ages, that chroniclers and reeves refer too, when they mention that the
corn did not germinate or was of stunted growth. Lost legumes crops are also commonly described in the manorial accounts;
peas and beans are drought sensitive (Pribyl, 2017; Pribyl and Cornes, 2019b; Stern, 2000; Titow, 1960, 1970). This form of
scarcity is also observed by the Venetian ambassador in early summer 1636, which followed on the dry 1635, and based on
his Italian background he anticipates a worsening of the scarcity due to the continuation of drought throughout summer 1636
– but he projected his Italian experiences on England and this expected dearth did not manifest, grain prices remained stable
after the harvest 1636, only the price for peas had been rising substantially since 1635 (Munro, 2008). As Merle had pointed
out, as long as one grain was sufficiently plentiful, and in drought this was wheat, there would be no famine. This mechanism
of exchanging one grain for another was also, what would contribute to ending famine in England in the seventeenth century,
when it became possible also after harvest failures linked to wet and cold summer seasons, to replace one grain with others;
when this substitution was possible the demographic effects of the subsistence crisis were minimized (Appleby, 1979).

A possible exception to the rule that 'Drought never bred dearth in England' must be considered: the year 1556. Not much is
known about the weather in spring and summer 1556, but the year played a role in massive rises in grain prices, the outbreak of
disease and a consequent extreme rise in mortality. Some compilations such as Lowe (1870) or Short (1749) mark it as a year of
drought, but it is not clear if they are basing their judgement on English or continental sources. On the continent the conditions

---

[3]Bodleian Library, Oxford, MS Digby 147, fols. 125-138.



in 1556 were indeed extremely warm and dry, e.g. in the vine harvest date series of the Vienna hospital 1556 was warmer than even 1540 (Buisman, 1998; Glaser, 2013; Kiss, 2018). Sources relating to the drought over the continent abound, but if the

spatial extension of the phenomenon included England can not be deduced from them. In a year when religious tensions ran high in England and the country's political and religious future was unsettled , few sources focus on the weather. Hence it is not surprising that some economic and social historians as Hoskins (1964) assume very wet conditions, as are classical in the development of dearth in England, while others stated a spring drought, though without giving the original evidence.

Jones et al. (1984) identify the summer season as dry based on Francis Blomefield's 'An essay towards a topographical

history of the County of Norfolk' (1806) who describes a dry summer during which in several places the turf caught fire and burnt for up to two weeks. Even though Blomefield is not a contemporary source for the mid-sixteenth century, he was using a manuscript from the Norwich archives of that time. Interestingly field and turf fires were also common across the North Sea in the Low Countries (Buisman, 1998). The last rainfall that can be traced for England in spring 1556 occurred on 21 March [Old Style] in Oxford at Cranmer's execution (Strype, 1694). Not much precipitation can have fallen after that, because on 16

June [Old Style] the Venetian ambassador to England did expected a 'risk of great sickness and yet greater famine than the last, owing to the heat and extraordinary drought of the season, as, contrary to the wont of this climate, and to the need of the soil, four months and upwards have passed without any rain to do good' (Brown, 1877). The situation did not improve and by 7 July [Old Style] 'intense heat' had joined the drought and 'processions are made continually' to relieve the situation (Brown, 1877). By mid-August another complaint about the heat, which now affected Queen Mary, was added (Brown, 1877). The

drought 1556 probably even stretch far north up to Scotland, where the authorities in Edinburgh commissioned a new well and the deepening of three existing wells in July (Marwick, 1871). Even though the drought was under-referenced in the official sources of the time, it comes up in Protestant propaganda at the beginning of the reign of Elizabeth I when a previous drought and dearth – undoubtedly in terms the drought a description of 1556 and in terms of the dearth the years following the harvests 1555 and 1556 – were laid at the feet of the godlessness of Mary's reign, even though the writer had spent those years

in exile on the continent, where the drought haunted catholic and protestant lands alike (Scholefield, 1842). In Kent it seems as if in the summer heat the harvest was early and the spring corn, mostly barley was too thin and of stunted growth in this year, so it could not be mown with short scythes. Possibly there were also field fires in Kent (see Table S1). In any case the harvest was poor and the grain prices rose to extreme heights afterwards (Munro, 2008). So indeed drought played a role in the dearth that ensued. However, as the Venetian ambassador indicated, famine reigned already before the harvest 1556 (Brown, 1877),

and prices had indeed been on a trajectory to famine levels since the harvest 1554 and reached them with the wet harvest 1555. Then in autumn 1555, at sowing time of the winter corn heavy rainfall and flooding set in and must have damaged the seed or delayed sowing (Bowden, 1985; Grafton, 1809; Holinshed, 1808; Hamilton, 1877). Hence when scarcity already was bad, the drought withered spring corn could not be substituted with wheat or rye and full famine ensued. This pattern is identified by Titow (1960) already for the Middle Ages: a wet autumn followed by a dry spring could cause scarcity, although a run

of wet conditions from autumn into spring/summer was more commonly linked to dearth and more severe harvest losses. In summer 1556 began a multi-annual phase of high levels of disease and mortality, which probably interrupted the harvesting, transporting and marketing of the grain further and could have contributed to raised grain prices.



## 5.2 Pastoral farming

In pastoral farming the impact of drought was more immediate that in agriculture: even a short drought, only lasting through
spring and early summer, could wreak havoc. During the study period livestock mainly depended on grass and hay and in
some regions on green fodder crops like vetches, so spring and summer rainfall were essential. Turnips which have a different
growing season, only appeared towards the very end of the study period (Overton, 1989). The growth of hay and herbage is
limited or even ceases under drought stress. In early modern times, between 1640 and 1740, wheat and hay prices have been
found to be generally inverted, that means years good for wheat – dry years – were bad for hay (Bowden, 1985). In the severe
drought of 1976 the grass growth in spring was s still adequate and potentially even higher in nutrition than normally, but
then regrowth after mid-June to July was disappointing (Carter, 1978). The restraint of feed then lead to a collapse of milk
production in dairy cattle herds in July and August, which took months to recover even with the return of rain (Carter, 1978).
Medieval manorial accounts mirror such conditions (Stern, 2000; Titow, 1960, 1970). Matthew Paris describes for 1252, how
grass rubbed between the hands turned to dust (see Table S1). In the Middle Ages sheeps' milk was also frequently used for
cheese making in England, and sheep fare better in prolonged dry weather than cattle (Carter, 1978). In the drought 1976 cattle
remained lean at the end of summer and both calves and lambs were underweight by September even with supplementary feed,
for calves this was sometimes a drastic difference in weight compared to a normal year (Carter, 1978). Drought stress appears
to have had an effect on the fertility of cows in the Middle Ages, which are often reported to have been barren in manorial
accounts during and shortly after drought periods (Pribyl, 2017). The capacity of lean livestock to survive winter is obviously
limited, but a lack of pasture and hay are even bigger factors in lowering winter survival rates (Bowden, 1985), hence livestock
needs to be sold before winter (Jones and Baker, 1964). Real desperation reigned when livestock was sold after winter (Jones
and Baker, 1964), as observed by Josselin for cattle in early 1652 during the drought 1651-53 (Macfarlane, 1976), indicating a
delayed onset of grass growth. A short term glut of livestock on the market for very low prices would be followed by a longer
period of high prices for the animals.
For the period 1200 to 1700 even more severe impacts are related for extreme droughts. Actual water shortage became a
danger for cattle. In particular cows with calves are in need of high amounts of water to maintain milk production (Carter,
1978), and Thomas Tusser (1812) states in the second half of the sixteenth century, that cattle needed to be watered daily in
summer. When water levels fell too much at pastures, cattle needed to be led to water, in the droughts 1326, 1540 and 1634
this was partly over distances of 3-12 miles (Anon., 1882; Hamilton, 1875; Whiteway, 1991). For summers 1384 and 1540 it
is reported that much cattle died due to a shortage of water (Hector and Harvey, 1982; Hamilton, 1877), although poisonous
weeds, that might have been consumed by the animals when no grass was left (Carter, 1978) and heat stress might also have
played a role in this raised cattle mortality. Heat stress, or potentially sun stroke, is attested to having killed people in 1473,
when 'in feld in harvist tyme men fylle downe sodanly' (see Table S1). In the worst case, the drought stress weakened the cattle
so far, that murrain could spread more easily as in 1252 and 1662 (Dobson, 1997; Luard, 1880). Drought before 1700 had thus
a high potential to disrupt pastoral farming, and the shortfall in livestock would cause a scarcity of dairy products and also
meat, which would impact especially the diet of the poor negatively by reducing their access to protein and fat. Additionally a





cattle murrain could reduced also the number of draught animals, which were in the medieval period still often oxen not horses, and hence have severe repercussions for transport and agriculture.

## 5.3   Health: Malaria, gastrointestinal disease and plague

The most immediate and severe risk of drought to life in England between 1200 and 1700 did, however, not operate through the food supply, but was a direct threat to health. In England crisis mortality in pre-industrial times was primarily connected to either subsistence crises caused by extreme rainfall levels or to epidemics, but summer warmth exerted an influence over the death rate also in non-crisis years. For the period 1665 to 1834 – the post-plague period in early modern England – when robust statistical data on mortality is available since the start of the parish registers in 1539, monthly death rates have been compared

to food prices and monthly temperature averages form the Central England Temperature series by Lee (1981). The analysis reveals that food prices exerted a greater influence upon mortality than temperature, even though this was the age, when famine was disappearing from England and the influence of prices on the death rate was waning over time. However, cold winters and warm summers did also increase mortality, the impact of summer heat was stronger than that of the winter cold. Lee calculated that a warming of 1°C in summer would raise the annual death rate by c. 4%. Most of the additional deaths would fall to late

summer and autumn, since the impact of summer heat on mortality was delayed by 1-2 months. Lee (1981) could not detect a significant influence of rainfall on mortality levels, but he had only access to annual rainfall totals, starting in 1727, and not only is early rainfall data problematic, but annual rainfall totals are not representative of summer precipitation. Other data help to identify periods of drought and heat that were marked by high mortality: records of heriot payments show a raised death rate amongst adults for areas mostly in eastern England from the mid-1320s to the early 1330s (Pribyl, 2017; Stone, 2014).

Postan and Titow (1959) in their study of heriots on the Winchester manors between 1245 (advanced to 1209 by Titow (1960)) and 1350 found a number of years, when mortality was high amongst villagers without an obvious connection to raised grain prices, and suspected drought induced epidemic disease due to poor sanitation, lack of and poor quality drinking water as a cause. This is evident for the high mortality following on the very dry and hot summer 1288,[4] which is also described in the Annals of Dunstable (Anon., 1866a; Postan and Titow, 1959; Titow, 1960). Drought was also present during the mortality

peaks 1236, 1300 and 1328 (Luard, 1876; Pribyl, 2017; Titow, 1960). It seems to have played a role in the raised death rate 1248 and in late 1272, although these years were also times of dearth (Anon., 1885; Postan and Titow, 1959; Titow, 1960).[5] Once the parish registers started, they immediately document the sharp rise in mortality during the drought summer 1540, when hot agues, dysentery and plague were afflicting the English (Wrigley and Schofield, 1981, see Table S1). Awareness of the dangers of hot dry summers existed in early modern times (Appleby, 1980; Bacon, 1670; Dobson, 1997), and these hot and

dry seasons did not influence the death rates by an influence on farming or food prices (Lee, 1981), but by directly contributing to the development of disease – as the case of 1540 demonstrates – often digestive disease and malaria.

---

[4]Mistyped as 1287 in Titow (1960).

[5]For the other years of raised mortality 1216, 1233, 1308, 1311 and 1342 there is either no documentary evidence on weather available (1215-16), or the evidence does not show drought periods as a potential primary driver of the mortality (Titow 1960; Symons, 1891; Stern 2000).





Medieval and early modern narrative sources also frequently mention higher death rates and unidentified diseases in conjunction with heat and drought. These are partly reported as 'fevers' or 'agues' – terms which also include, but are not limited to, malaria – as in 1222, 1242, 1252, 1285, 1288 and 1305 (Anon., 1866a; Coggeshall, 1875; Hewlett, 1889; Luard, 1877, 1880;
Postan and Rishanger, 1865; Titow, 1959). The following period is dominated by a focus on plague epidemics, but from the second half of the fifteenth century onwards non-plague disease is again reported for the summer droughts 1464, 1473, 1540, 1612, 1616, 1653, 1657-58, 1669-70 and 1679-80 (Anon., 1880a; Dobson, 1997; Fabyan, 1811; Halliwell, 1839; Hamilton, 1875; Pribyl, 2017). Occasionally 'quartan agues' or 'fevers' are described in the records, i.e. fevers at three-day intervals, which raises the chances of these references to be indeed reports of malaria, in 1222 the source comes from eastern Essex, in
1242 from the well-informed Matthew Paris at St Albans, in 1557-58 'quartan fevers' seemed to have been part of the mixture of disease haunting the English during and after the famine caused by the bad harvests 1555 and 1556, although influenza appears to have been the main culprit of the enormous death rates (Appleby, 1980; Creighton, 1891; Fabyan, 1811; Hamilton, 1877; Holinshed, 1808; Stow and Howes, 1631). For 1252 Matthew Paris writes, how the heat did also continue into the night and flies, fleas and other insects made life for men and beast miserable (see Table S1), possibly this description includes the
activity of the nocturnal *Anopheles* mosquito which acts as a transmission vector for malaria. In 1464 then, near Ely 'great poisonous and horned flies' killed their victims quickly (see Table S1); whereas this seems to refer to horseflies, the stagnant waters of that region – increased by the falling water levels in the drought – must have also been a breeding ground for other insects. Due to the geographical restrictions of the *Anopheles* mosquito which acted as a transmission vector for malaria, this disease was more likely to occur in low lying – and in the study period still largely undrained - wetlands as around the Isle of
Ely or the marshlands on the eastern and southern coasts and in the Thames estuary. Indeed many of the references to disease in drought summers mentioned above come from men of close geographical proximity to marshes, f.e. for 1222 by Ralph de Coggeshall (eastern Essex) or from well informed men who also showed a repeated interest in marshland as for 1242 by Matthew Paris (Hertfordshire). In the early modern period hot summers caused a sharp increase in mortality in south-eastern England; this summer-autumn mortality occurred in upland as well as wetlands, but was much more pronounced in the latter
(Dobson, 1997). Also in this regional setting, the variability in summer-autumn mortality was more strongly connected to weather, in this case temperature, than mortality at other times of the year (Dobson, 1997).

In the period 1661-1800 the correlation between July-August temperature and the burials in the year following was high in the marshlands; cool summers suppressed the death rate in the subsequent months (Dobson, 1997). Malaria was endemic in the marshes; by the nineteenth century infection resulted rather in illness than death for local adults, but children had more intense
attacks and were at higher risk. In the early modern period the mean death rate, however, was 30-50% higher in the marshland than elsewhere in south-eastern England (Dobson, 1997). Malaria, however, is unlikely to have affected the whole of England even in warm summers, due to the spatial range of the *Anopheles* mosquito being limited to wet lowlands.

Gastrointestinal disease was another cause for increased mortality levels in and after drought summers; it affected particularly young children after weaning. The various forms of transmission, via contaminated water and food, or via flies, were all
enhanced during dry and warm summers, when water levels were low and as described for 1252 and 1464 insect populations increased dramatically, especially in areas of standing water. Water could be made unsafe to drink by bacteria or algae, in 1409



f.e. 'blood is seen to gush forward from wells in divers parts of England, and in consequence many died of dysentery' (Haydon, 1863, p. 415, translation by the author). The 'bloody flux' or 'laskes' [dysentery] were part of the diseases in 1473 and 1540. Gastric problems were also present in 1612, 1624, 1669-71, 1678-80 and possibly in 1616 and 1657-58 (Dobson, 1997). Again

the marshes were overly affected, because there and in the towns the access to clean drinking water was particularly difficult (Dobson, 1997) and flies found plenty of breeding grounds. The relationship between hot and dry conditions and increased (childhood) mortality due to digestive disease persisted well into the nineteenth century (Dobson, 1997).

As serious a threat as various fevers like malaria and gastric infections posed, the disease that really raised alarm amongst the populace was plague. In the First Pestilence in the mid-fourteenth century more than a third of the English died, another

10% fell victim to the Second Pestilence 1361-62. Large scale – in terms of geographical coverage and death toll – plague outbreaks marked the remaining fourteenth century, in the fifteenth century the outbreaks assumed a more regional and partly urban character and appear somewhat more scattered, though very severe outbreaks returned in the second half of the century; plague waves occurred every 5-12 years. Around 1500 Scotland and England began to implement measures intended to hinder the development and spread of the plague, such as quarantining, isolation, notification and travel bans. Up to about 1560 there

seems to be only a comparatively small number of severe outbreaks affecting the country, a picture that turned during the last decades of the century. Plague remained a serious threat until the last wave in 1665-66, in England major plague waves caused higher death rates than famine in the sixteenth and seventeenth century (Appleby, 1980). Throughout the period plague was often present in London for long periods, although death rates due to plague during these phases remained generally low, apart from crisis years that sometimes overlapped with outbreaks in other parts of England.

When considering the relationship of the disease to meteorological factors, it is useful to focus on large -scale outbreaks. The Central England Temperature series starts in 1659, and hence hardly overlaps with the era of plague, and no yearly or seasonally resolved reliable temperature or precipitation index is available for England between 1350 and 1700, so that the van Engelen (2001) temperature index from the Low Countries needs to be employed for the analysis; it can be supported by evidence from England. During the Middle Ages a specific weather pattern was associated with major plague outbreaks. Years

of high plague mortality were generally marked by warm and dry summers, in any case these summers were warmer and drier than the previous summers which were on the cooler or wetter side, but never extremely cold or wet. The winter preceding the high plague mortality was not a hard winter, it could be of mild conditions or medium severity. Extremely cold winters were detrimental to plague, not only did they not precede severe plague years, they could also mark the end of a plague wave. Only a minority of plague outbreaks can be associated with cool and wet summer years, failed harvests and dearth, the most

important examples being the First Pestilence 1348-49, the Third Pestilence in 1369 and the plague wave just after the failed harvests in the second half of the 1430s (Pribyl, 2017). In fact up to the end of the Middle Ages the relationship between warm and dry summers, which were at least noticeable warmer and drier than previous summers and did not follow on a cold winter or an extremely cold and wet summer, and major plague outbreaks was strong, if 5-12 years since the last outbreak had passed. Only towards the end of the fifteenth century appear such meteorological conditions without a plague wave (Pribyl, 2017).

In the subsequent centuries this relationship weakened, but the more substantial outbreaks of 1513, 1532, 1536, 1563, 1575, 1578, 1592-93 and 1590-91 (Southwest), and almost all in the seventeenth century, 1603, 1609, 1635-36 and 1665-66, again



coincided with the aforementioned weather pattern; early modern outbreaks can sometimes be traced back to the infection taking root in a port town. The plague outbreak 1624-25 started under the standard conditions, but 1625 itself was cool and wet. However, in the sixteenth and seventeenth centuries other years offered the meteorological conditions for plague outbreaks

and came after a 5-12 year interval since the last major outbreak, but saw no major plague wave. It is outside the scope of this paper to speculate on the reasons behind this weakening of the relationship, if this was due human intervention, a change in the epidemiology of plague or increasing immunity. Again the high mortality due to plague in hot and dry years was not due to dearth – in London there was no link between plague mortality and bread prices in the sixteenth and seventeenth centuries (Appleby, 1975), nor were most major plague outbreaks in the Middle Ages linked to high food prices (Pribyl, 2017). The link

between hot and dry summers and plague was known to contemporaries (Pribyl, 2017), and in 1670 Francis Bacon could state in *Sylva Sylvarum*, that 'In England [...], many times, there have been great Plagues in dry years' (Bacon, 1670, p. 84).

## 5.4 Fire risk

A further risk in times of persistent dry weather was the increased chance of fires, particularly during multi-annual droughts. As London fell victim to the flames in the dry summer of 1666, so other town fires occurred during droughts, as in 1288 and 1303

in Boston and in 1326, when Royston, Wandsworth and the abbey of Croxden were consumed by fire (Anon., 1996; Aungier, 1844; Ellis, 1859, Rishanger, 1865). In 1389 and 1393 fires following on thunderstorms destroyed villages in Kent, in spring 1394 fires were widespread in eastern England (Hector and Harvey, 1982;Anon., 1866b). When summers were extremely dry and hot, turf, fen or fields could also catch fire, as in 1556 in Norfolk (Blomefield, 1806) and possibly also in Kent (Beale, 1998). Norfolk also saw a sharp rise in field fires in the dry and warm summer of 2018.

## 5.5 Water supply

As mentioned above, low water levels threatened the health of people and livestock by raising the risks of contamination to water sources or dehydration particularly in marshland and in towns. London was using the Thames over long periods as a source of drinking water, and when the fresh water supply dwindled, then salt water would intrude towards the west of London Bridge further than normally. In 1326 Londoners had to content themselves with salty ale (Aungier, 1844). The Thames was

also very low in 1540 and 1592 (Stow and Howes, 1631). When the Thames dropped too low, then the barges between Oxford and London could not operate as in 1634, 1675-76 and 1685 (Clark 1892, 1894; Jones et al., 1984). Water transport elsewhere was also hindered, as 1326 around the Isle of Ely (Anon., 1882). Springs drying up was common as is indicated in 1236, 1241, 1252, 1326, 1384, 1540, 1619 (Anon., 1882; Aungier, 1844; Clyn, 1849; Grafton, 1809; Hamilton, 1875; Hall, 1809; Hector and Harvey, 1982; Holinshed, 1808; Jones et al., 1984; Luard 1877, 1880; Stow and Howes, 1631; Titow, 1960). The

marshland itself also tended to be subjected to falling water levels as it is stated for 1236, 1241 and 1352 (Luard 1876, 1877; Lumby, 1895). Falling river levels also reduced the availability of water power and affected industrial and grain mills. Explicit evidence for mills stopping to work has been given by Matthew Paris (Luard 1876, 1877, 1880) for 1236, 1241 and 1253, the Pipe Rolls of the Bishopric of Winchester mention mills standing idle for a lack of water for 1248, 1268, 1327, 1336, 1337, and 1340 (Titow, 1960), and a mill also lacked water in 1435-36 (prob. 1435) and 1619 (Jones et al., 1984; Langdon, 2004).





When the mills could not grind the corn, a flour shortages would ensue, even if the harvest had been good and prices not raised. In 1540 'people woulde haue giue one bushel [of grain] for the grindyng of another' (see Table S1), so the price for grinding was very high. Unsurprisingly disputes about water courses, their blockage or diversion, and mills often flared up in dry years.

### 5.6  Flash flooding

Paradoxically, droughts were also associated with flooding: once rainfall returned, the dried up, hardened soil often could not
absorb the water at sufficient speed and flash flooding was the consequence. This has been described for 1253 and 1393 (Anon., 1866b; Luard, 1880; Lumby, 1882; Riley, 1864).

## 6  Adaptation measures: water supply, marshland drainage and fodder crops

Since the severity of drought impacts was evident, not only individuals but also communities and institutions were adapting to the risk. Most pressing was the lack of water for people, industry and fire fighting, and hence improvements of water
infrastructure are often mentioned for drought years. In 1240 the Countess of Devon brought water over five miles to her manor of Tiverton (Brooks and Glasspoole, 1928). In King's Lynn a cistern fed by pipes was installed in 1386 after at least two dry summers, and in 1474 Norwich intended to construct two new wells after the extremely warm and dry summer 1473 (Rawcliffe, 2013; Pribyl, 2017; Halliwell, 1839). The drought 1556 seems to have stretched up to Scotland, where Edinburgh deepened three of its existing wells and undertook the construction of a new one (Marwick, 1871). The first water from springs
outside Edinburgh arrived in the city in the drought summer of 1676. After recognizing the need for a public water supply since more than a decade, Plymouth built Drake's Leat, which brought water from Dartmoor, in the multi-annual drought 1590-91 (Brooks and Glasspoole, 1928) and Norwich replaced its well with a pump in 1685, the second dry summer half year in a sequence (Blomefield, 1806). In London, the first water conduit is recorded for a drought year, 1236, and the New River was begun in the 1609, a year marked by a dry spring and summer, the King became directly involved to help countering difficulties
of the project in the severely dry 1612 following on the drought in summer 1611, and the New River was completed in 1613. In the drought year 1616, which again followed another severely dry summer half year, the New River Company was formed (Stow and Howes, 1631; Tomory, 2017). The number of customers of the New River water doubled during the drought summer 1615 and increased by another third in the dry 1616, but then growth slowed and stagnation and recession marked the 1620s; in the dry summers between 1630 and 1638 the number of customers doubled again (Tomory, 2017). Attempts were also made to
offset the lack of water power by harnessing other sources of power, even though this is hardly traceable: in 1325-26, a multi-annual drought, a windmill was erected in East Knoyle, Wiltshire, and in 1408-09, when the latter year was dry, a horse-mill was established in Ivinghoe, Buckinghamshire (Langdon, 2004; Pribyl, 2017).

Droughts had the advantage of facilitating the drainage of the marshland; drained marshland would increase the available agricultural land. Already in the Middle Ages the draining of fenland fell to times drier than average (Hallam, 1965). The ob-
vious advantage of low water levels for drainage projects was also utilized, when the seventeenth-century large-scale drainage was undertaken. In 1605 the Commissioner of Sewers for Norfolk noted that the long lasting dry summer weather offered a





good opportunity to act upon the royal wishes for draining the fenland (Morgan et al., 2010). The decision to drain the Great Fen in Cambridgeshire and Norfolk was taken in 1630, a dry summer half year, and then carried out during the generally dry 1630s. A second phase fell to the dry 1650s (Albright Knittl, 2007). As with the installation of an improved public water supply in growing towns, the wish to drain the marshland was long-standing, but then dry conditions either forced the authorities into action or offered an easy opportunity for a realisation of plans.

The seventeenth century saw the beginning of the Agricultural Revolution. Not only was the sown acreage expanded and so grain output raised, but farming methods and productivity also improved. The cultivation of turnips in the second half of the seventeenth century was part of that change. Turnips served largely as animal feed and allowed higher levels of animal husbandry in areas, where pasture or hay were in short supply. However, considering the high frequency of drought summers in the 1660s, 1670s and 1680s – the period when turnips were introduced – the root vegetable also allowed to offset the effects of drought on pasture and hay. Tubers are sown as late as August, when the negative effects of dry weather on grass and the hay crop were obvious, and turnips could hence supply extra winter feed in years (Overton, 1989). Another way to ensure a supply of grass – although not in defiance of a summer drought, but as a way to ensure an early fodder supply in spring – was the irrigation of meadow grass in the form of catchwork meadows or floated meadows, the latter system was elaborate and first described in 1610, but had come into operation in years before 1600, the first method is even older (Jones and Baker, 1964).

As the drainage of the fenland raised the supply of arable land and pasture, contemporaries also considered the drainage efforts as a measure to improve public health. The vapours and smells of marshland were perceived as miasma, which was thought to be the cause of disease. However, the drainage of the marshes did indeed reduce the breeding grounds for the *Anopheles* mosquito and hence did contribute to improved health, although the value of seventeenth-century drainage in this respect remains doubtful (Dobson, 1997). Elsewhere efforts for a better water supply, and for sanitation and hygiene, such as improved street cleaning, also must have helped to reduce the effects of dry hot weather particularly on the occurrence of enteric fevers and maybe even plague.

## 7 Conclusions

Spring-summer droughts have been a feature of the climate in southern and eastern England over the centuries. Often the years marked by dry conditions clustered. Severe droughts are known to have occurred in 1252, 1325-26, 1361, 1464, 1540, 1590-91, 1636, 1676 and 1685. In medieval and early modern times, agricultural drought was rarely involved in the development of famines, because the wheat crop was resilient to warm and dry conditions. The pastoral sector, however, came under immediate and potentially severe threat during spring-summer droughts. The hydrological aspects of drought impacted the water supply of people and livestock, and also limited river transport and the energy supply of flour and industrial mills. Towns before the widespread use of stone in buildings had a raised risk of fires spreading through the settlement in dry years. Drought – often going hand in hand with warmth in spring and summer – could severely affect health: outbreaks of malaria, gastrointestinal disease and plague often fell to drought years.





*Author contributions.* Written by K.Pribyl.

*Competing interests.* No competing interests.





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



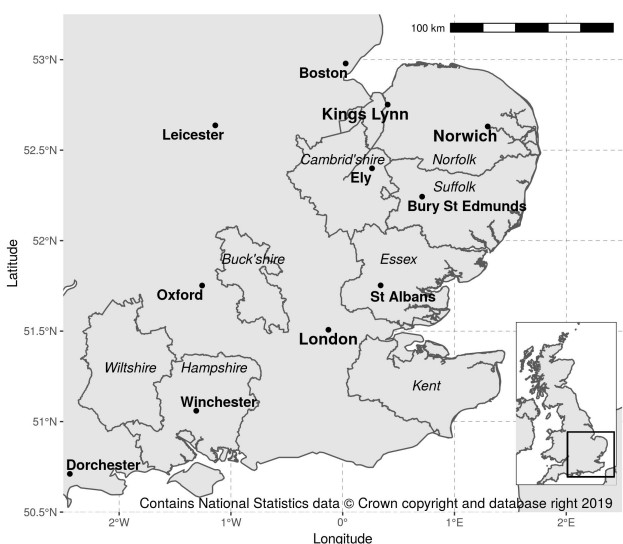

**Figure 1.** Map showing regions and cities in England described in the text. Counties indicate modern day administrative boundaries.

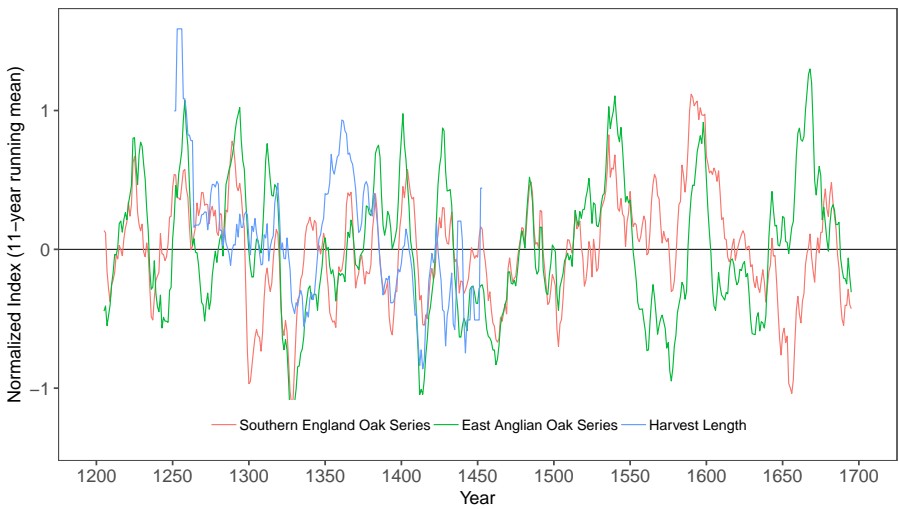

**Figure 2.** Time series of the tree-ring based precipitation reconstructions by Wilson et al. (2013) for southeast England and by Cooper et al. (2012) for East Anglia, alongside the precipitation index for East Anglia by Pribyl (2017). Series are normalized over the period 1200 to 1700.