# Peer review of "A survey of the impact of summer droughts in southern and eastern England, 1200-1700"

_Climate of the Past, 2019_

## Referee Comment (RC1) · Neil Macdonald (Referee) · 21 Oct 2019

This is an interesting and enjoyable paper that presents new research on long-term droughts in South East England. I have attached an annotated version of the manuscript with suggested edits both in terms of language but also suggested additional considerations. I hope the author finds these comments helpful, the paper would also benefit from a detailed proof read, as in several places sentence structure / length needs reviewing.

Key points to address: Consider renaming the paper . . . .summer droughts in South East England. . . as the focus is on SE, as acknowledged.

I would encourage the author to consider the full breadth of drought reconstruction

work undertaken for England, with several papers not mentioned, some of which cover the area of study (e.g. Spraggs et al., 2015 and Todd et at., 2013). This will help contextualise the work better.

The section on 1540 would benefit from strengthening with contemporary accounts, these from experience are limited, indeed only accounts are from London used (e.g. documented in Wetter 2013), an additional source for London may be present in the Cramner register, concerning call for prayers from Henry VIII. No evidence from outside London (SE) from a contemporary source?

Take care with statements on droughts in particular climates, drought is equally liable to occur in SE as NW England, however the impacts may differ.

Rename the 'Drought over the centuries' section, as this seems to be more about source material through the centuries than drought over the centuries.

I think some reconsideration of Section 6 adaptation measures is needed, some of this section addresses changes, but it is not clear how this differentiates from changes that were already being made/ how droughts drove these adaptations, rather than just coinciding with these changes. If you are making these links they need to be explicit, e.g. London New River first proposed in 1602, so not linked to drought of 1609 when first opened.

The author may like to add a sentence on the utility of this paper on better understanding droughts today? Value of such information in water resource plans.

I have suggested additional references on the annotated manuscript. The author may also find the following of interest:

Brázdil, R., Kiss, A., Luterbacher, J., Nash, D.J., ÅŸezníčková, L., 2018. Documentary data and the study of past droughts: a global state of the art. Clim. Past 14, 1915–1960. doi: 10.5194/cp-14-1915-2018

Sangster, H., Jones C., Macdonald N. (2018) The co-evolution of historical source

materials in the geophysical, hydrological and meteorological sciences: Learning from the past moving forward, Progress in Physical Geography, 42(1): 61-82 doi: 10.1177/0309133317744738

Neil Macdonald University of Liverpool

Please also note the supplement to this comment:
https://www.clim-past-discuss.net/cp-2019-116/cp-2019-116-RC1-supplement.pdf

———————————————————

[Figure]

**Supplement:**

[revised manuscript text omitted]

---

## Referee Comment (RC2) · Alexander Hall (Referee) · 4 Nov 2019

This is an interesting and timely paper, and I recommend that following minor revisions it should be accepted for publication.

In keeping with the comments from reviewer 1, I think the author should re-frame the paper to be explicitly about S.E. England, in addition to amending the title to reflect this focus, clearer framing in the introduction and section 2 could be added.

Given the centrality of the sources to the paper, would the author be able to include Table S1 in-line in the text of the paper? If not in full, an amended or overview version would help the reader interested more in the historical context of the sources in gaining a quick overview of the kind of narrative sources used in the study.

[Figure]

In section 6 I would like the author to add a paragraph reflecting on the ability to assess the direct causality of adaptation measures, were they measures triggered solely by drought events, were they catalysed by meteorological conditions etc.

As per the corrections and comments on the attached PDF the author needs to give the manuscript a thorough proof read and review prior to resubmitting. In addition to minor amends on typographical errors, the author should ensure all sentences and paragraphs are clear, avoiding long and unwieldy sentence structures where possible.

**A survey of the impacts of summer droughts in England, 1200-1700**

Kathleen Pribyl[1,2]

[1]Climatic Research Unit, School of Environmental Sciences, University of East Anglia, UK
[2]Oeschger Centre for Climate Change Research, University of Bern, Switzerland
**Correspondence:** Kathleen Pribyl (k.pribyl@uea.ac.uk)

**Abstract.** Droughts pose a climatic hazard that can have a profound impacts on past societies. Using documentary sources, this paper studies the occurrence and impacts of spring-summer droughts in pre-industrial England from 1200 to 1700. The types of records, source availability and changes in record keeping over time are described, and an overview of droughts in those 500 years is provided. The focus lies on a structural survey over the drought impacts most relevant to human livelihood.

5  This includes the agricultural and pastoral sectors of agrarian production, health, the fire risk to settlements and the drop in water levels or dwindling of water supplies. Whereas due to the specific characteristics of wheat cultivation in medieval and early modern England, the grain production was comparatively resilient to drought, livestock farming was under threat when rainfall fell noticeably below average. The most important problem in warm and dry summers, however, was the risk to health. Partly steeply raised mortality levels were associated with these conditions during the study period, because malaria, gastrointestinal

10  disease and plague showed an affinity to heat and drought. Adaptation strategies to reduce the stress posed by summer droughts are included in the study.

**1 Introduction**

The occurrence of drought across the British Isles has attracted the attention of meteorologists since the nineteenth century. G.J. Symons' work on hydrometeorology is well known (1887). He was not only studying the rainfall patterns of his own time by

15  establishing an extensive rain gauge network across the British Isles (Jones, et al. 2007), but also investigated the occurrences of droughts over the past on the occasion of the 1887 drought (Symons, 1887). Brooks and Glasspoole (1928) based their work on past droughts on the catalogue of Symons. After them most research has been done focused on the instrumental period, as by Jones et al. (1997), Briffa et al. (2009), Cole and Marsh (2006) and Marsh et al. (2007). Recently Ireland's drought history was investigated by Wilby et al. (2016), Murphy et al. (2017) and Noone et al. (2017). Some of these works also

20  include documentary data which is used for an evaluation of the drought conditions and impact research (Cole and Marsh, 2006; Murphy et al., 2017; Noone et al., 2017).

Drought in the pre-industrial period has received comparatively little attention. Using information in the form of direct weather references or proxy data in documentary sources, drought occurrence in England was included in the studies by Jones et al. (1984), Ogilvie and Farmer (1997), Pribyl (2017) and Pribyl and Cornes (2019a,b), and for Ireland by Dooge (1985). Over

25  the last decades a new source for the study of droughts on the British Isles extending back to Antiquity has become available: precipitation reconstructions based on tree-ring data (Cooper et al., 2012; Rinne et al., 2013; Wilson et al., 2013). These data

**Fig. 1.**

---

## Author Comment (AC1) · 27 Jan 2020

Review 1 (Neil Macdonald)

This is an interesting and enjoyable paper that presents new research on long-term droughts in South East England. I have attached an annotated version of the manuscript with suggested edits both in terms of language but also suggested additional considerations. I hope the author finds these comments helpful, the paper would also benefit from a detailed proof read, as in several places sentence structure / length needs reviewing.

Response: I would like to thank the reviewer for taking the time for providing the annotated pdf. I will address the suggested changes in the revised paper.

Key points to address: Consider renaming the paper....summer droughts in South East England... as the focus is on SE, as acknowledged.

Response: The title of the paper will be changed to 'southern and eastern England'.

I would encourage the author to consider the full breadth of drought reconstruction work undertaken for England, with several papers not mentioned, some of which cover the area of study (e.g. Spraggs et al., 2015 and Todd et at., 2013). This will help contextualise the work better.

Response: I thank the reviewer for that suggestion and I will include reference to these articles in the introduction.

The section on 1540 would benefit from strengthening with contemporary accounts, these from experience are limited, indeed only accounts are from London used (e.g. documented in Wetter 2013), an additional source for London may be present in the Cramner register, concerning call for prayers from Henry VIII. No evidence from outside London (SE) from a contemporary source?

Response: The 1540 event and available printed sources are described in detail in Pribyl, K., Cornes, R., Drought in medieval and early modern England. Part 1. The evidence, Weather (2020). The sources provided there include the commissions for prayer by the king. The information listed in Lowe's Natural phenomena and chronology of the seasons' (1870) as coming from outside London is misplaced and misdated as pointed out in the abovementioned paper. The citation of this paper will be moved from the footnote relating to 1540 into the relevant text paragraph.

Take care with statements on droughts in particular climates, drought is equally liable to occur in SE as NW England, however the impacts may differ.

Response: I note that the reviewer has highlighted such sections in the annotated manuscript and I will change sections as indicated.

Rename the 'Drought over the centuries' section, as this seems to be more about source material through the centuries than drought over the centuries.

Response: This section aims to give a general overview of droughts per century and their documentary basis; I have changed the title to 'An overview of droughts 1200-1700'

I think some reconsideration of Section 6 adaptation measures is needed, some of this section addresses changes, but it is not clear how this differentiates from changes that were already being made/ how droughts drove these adaptations, rather than just

coinciding with these changes. If you are making these links they need to be explicit, e.g. London New River first proposed in 1602, so not linked to drought of 1609 when first opened.

Response: I will add a clarifying phrase, that – as in the Plymouth example – the wish for an improvement of the water supply might have been long standing, but drought years raised the need for better water supplies suddenly and decisively, and hence likely triggered the execution of planned costly infrastructure projects. The project of the New River lingered around for years before the project was begun in the drought year 1609, and was then pursued over the largely dry period – with royal involvement during the driest part - until it's completion in 1613. It is not unreasonable to assume that existing aspirations to improve the water supply in urban settlements due to a strain created by the growth of population or industry received a considerable impetus, when drought years endangered or diminished the availability of water.

The author may like to add a sentence on the utility of this paper on better understanding droughts today? Value of such information in water resource plans.

Response: I will add a sentence in relation to the utility of establishing a pre-industrial baseline of droughts in England. However, I feel that incorporating information about the use of the historical information for modern planning would deserve a detailed approach that is not possible in this paper.

I have suggested additional references on the annotated manuscript.

Response: I have already reviewed these references and will add as appropriate.

---

## Author Comment (AC2) · 27 Jan 2020

Review 2: Alexander Hall (Referee)

This is an interesting and timely paper, and I recommend that following minor revisions it should be accepted for publication.

Response: I thank the reviewer for the helpful comments and suggestions.

In keeping with the comments from reviewer 1, I think the author should re-frame the paper to be explicitly about S.E. England, in addition to amending the title to reflect this focus, clearer framing in the introduction and section 2 could be added.

Response: The title of the paper will be changed to focus on southern and eastern England. The regional focus is already highlighted at the start of section 2.

Given the centrality of the sources to the paper, would the author be able to include Table S1 in-line in the text of the paper? If not in full, an amended or overview version would help the reader interested more in the historical context of the sources in gaining a quick overview of the kind of narrative sources used in the study.

Response: The paper is already very long. Due to the constraints on length and the generally more natural-science oriented audience of the journal, long source citations are allocated to the supplementary information.

In section 6 I would like the author to add a paragraph reflecting on the ability to assess the direct causality of adaptation measures, were they measures triggered solely by drought events, were they catalysed by meteorological conditions etc.

Response: See response to reviewer 1.

As per the corrections and comments on the attached PDF the author needs to give the manuscript a thorough proof read and review prior to resubmitting. In addition to minor amends on typographical errors, the author should ensure all sentences and paragraphs are clear, avoiding long and unwieldy sentence structures where possible.

Response: In accordance with the suggestions by reviewer 1 the manuscript will be amended as necessary.

---

## Author Response (AR2)

A survey of the impact of summer droughts in southern and eastern England, 1200-1700

Kathleen Pribyl[1,2]

[1] Climatic Research Unit, School of Environmental Sciences, University of East Anglia, UK
[2] Oeschger Centre for Climate Change Research, University of Bern, Switzerland
Correspondence:  Kathleen Pribyl (k.pribyl@uea.ac.uk)

**Editor's comments**

I would like to thank the editor for his suggested changes. I have amended the manuscript as indicated in the attached document.

[revised manuscript text omitted]